# A Multiplicative Calculus Approach to Solve Applied Nonlinear Models

**Gurjeet Singh [1], Sonia Bhalla [1,]* and Ramandeep Behl [2]**

1   Department of Mathematics, Chandigarh University, Gharuan, Mohali 140413, Punjab, India
2   Mathematical Modelling and Applied Computation Research Group (MMAC), Department of Mathematics, King Abdulaziz University, P.O. Box 80203, Jeddah 21589, Saudi Arabia
*   Correspondence: soniamaths5@gmail.com

**Abstract:** Problems such as population growth, continuous stirred tank reactor (CSTR), and ideal gas have been studied over the last four decades in the fields of medical science, engineering, and applied science, respectively. Some of the main motivations were to understand the pattern of such issues and how to obtain the solution to them. With the help of applied mathematics, these problems can be converted or modeled by nonlinear expressions with similar properties. Then, the required solution can be obtained by means of iterative techniques. In this manuscript, we propose a new iterative scheme for computing multiple roots (without prior knowledge of multiplicity $m$) based on multiplicative calculus rather than standard calculus. The structure of our scheme stands on the well-known Schröder method and also retains the same convergence order. Some numerical examples are tested to find the roots of nonlinear equations, and results are found to be competent compared with ordinary derivative methods. Finally, the new scheme is also analyzed by the basin of attractions that also supports the theoretical aspects.

**Keywords:** multiplicative derivative; nonlinear equations; Schröder method; order of convergence

**MSC:** 65H05; 65G99

## 1. Introduction

In the seventeenth century, Newton and Leibnitz created the differential and integral calculus concept based on subtraction and addition operations. Later in the 1970s, Grossman and Katz [1] developed a different definition of differential and integral calculus that utilized the multiplication and division operation instead of addition and subtraction. This definition of differential and integral calculus is named multiplicative calculus. In 2008, Bashirov et al. [2] contributed to multiplicative calculus and its applications. After this, some authors worked on some applications of multiplicative calculus in different areas such as biology [3], science and finance [4], biomedical sciences [5], economic growth [6], etc. From the above discussion, we can say that the multiplicative calculus approach has an important role in the field of applied sciences [7–17].

In 2016 and 2020, Özyzpici et al. [18] and Ali Özyzpici [19] suggested a new way to solve nonlinear equations with the help of a multiplicative calculus approach (MCA). The numerical results of these methods [18,19] have been found to be much better compared with iterative techniques with a standard calculus approach. In these studies [18,19], researchers focused only on the simple root of nonlinear equations. They did not discuss multiple roots because finding the multiple roots of nonlinear expressions is a more complicated and challenging task compared to simple roots. Retaining the same convergence order and lengthy, complicated calculations, the complex body structure of the iterative method and computational efficiency are also other reasons. In addition, most of the iterative methods for multiple roots required prior knowledge of multiplicity $m$, which is not practically possible to obtain in advance. According to our best knowledge, we have

no iterative method based on the multiplicative calculus approach for multiple roots of nonlinear equations in the available literature.

While keeping these things in mind, we propose a new iterative technique for multiple roots with unknown multiplicity based on MCA. According to our best knowledge, we are the first to report such a scheme with a multiplicative calculus approach that can handle multiple roots. In addition, our scheme does not require prior knowledge of multiplicity $m$. The structure of our scheme stands on the well-known Schröder method. We compare our scheme with existing methods on the basis of absolute error difference between two consecutive iterations, order of convergence, number of iterations, CPU timing, the graphs of absolute errors, and bar graphs. We found that our methods perform much better in all ways of comparison. Finally, we study the basin of attraction of our method, which also supports the numerical results.

The details of the paper are as follows: Section 2 states the proposed multiplicative method; Section 3 represents the convergence analysis of suggested methods; Section 4 demonstrates the experimental work of newly constructed schemes; Section 5 is devoted to the graphical analysis of new methods; the last Section 6 depicts the concluding remarks.

*1.1. Some Basic Terminologies*

**Definition 1** ([2])**.** *The nonlinear function $g : \Omega \subset \mathbb{R} \to \mathbb{R}$ is multiplicative differentiable ($g^*$) at $x$ or on $\Omega$ if it is positive and differentiable at $x$ or on $\Omega$, and it is defined as*

$$g^*(x) = \frac{d^* g}{dx} = \lim_{h \to 0} \left( \frac{g(x+h)}{g(x)} \right)^{\frac{1}{h}},$$

$$g^*(x) = \lim_{h \to 0} (\triangle^* g)^{\frac{1}{h}} = e^{\frac{g'(x)}{g(x)}},$$

$$= e^{(ln \circ g)'(x)}. \tag{1}$$

In a similar pattern, the higher-order multiplicative derivative is defined as

$$g^{**}(x) = e^{(ln \circ g^*)'(x)} = e^{(ln \circ g)''(x)}, \tag{2}$$

and, more generally,

$$g^{*(n)}(x) = e^{(ln \circ g)^{(n)}(x)}, n = 0, 1, 2, \ldots \tag{3}$$

where $(ln \circ g) = ln(g(x))$. Note in Equation (3) that $n = 0$ means no multiplicative derivative and it depicts the original function $g(x) = 1$.

*1.2. Some Results on Multiplicative Differentiation*

Consider $g$ and $h$ to be multiplicative differentiable and $\psi$ to be ordinary differentiable functions. Let $c$ be a positive constant; then, we have

1.  $(c)^* = 1$
2.  $(cg)^*(x) = g^*(x)$
3.  $(g \circ h)^*(x) = g^*(x) h^*(x)$
4.  $(\frac{g}{h})^*(x) = \frac{g^*(x)}{h^*(x)}$
5.  $(g^\psi)^*(x) = g^*(x)^{\psi(x)} \cdot g(x)^{\psi'(x)}$
6.  $(g \circ \psi)^*(x) = g^* \psi(x)^{\psi'(x)}$

**Definition 2.** *Suppose $g : \Omega \subset \mathbb{R} \to \mathbb{R}^+$ is a positive nonlinear function. Then, the multiplicative nonlinear equation is defined as*

$$g(x) = 1. \tag{4}$$

**Theorem 1** ([20]). *Let $g : \Omega \to \mathbb{R}$ be $(n+1)$ times multiplicative differentiable in an open interval $\Omega$. Therefore, for any $x, x + a \in \Omega, \exists$ a number $\eta \in (0,1)$ such that*

$$g(x + a) = \prod_{l=0}^{n} \left( g^{*(l)}(x) \right)^{\frac{a^l}{l!}} \left( g^{*(n+1)}(x + \eta a) \right)^{\frac{a^{n+1}}{(n+1)!}}. \tag{5}$$

## 2. Proposed Schemes

Here, we consider the well-known Schröder Method defined as

$$x_{k+1} = x_k - \frac{g(x_k)g'(x_k)}{(g'(x_k))^2 - g(x_k)g''(x_k)}, \quad \forall k = 0, 1, 2, \cdots. \tag{6}$$

We replace the ordinary derivative $g'(x_k)$ and $g''(x_k)$ of the function $g(x_k)$ with multiplicative derivative $ln(g^*(x_k))$ and $ln(g^{**}(x_k))$ in the method (6) and obtain the following iterative method to solve the nonlinear equation:

Multiplicative Schröder Method (MSM)

$$x_{k+1} = x_k - \frac{ln(g(x_k))ln(g^*(x_k))}{(ln(g^*(x_k)))^2 - ln(g(x_k))ln(g^{**}(x_k))}, \quad \forall k = 0, 1, 2, \cdots. \tag{7}$$

## 3. Convergence Analysis

**Theorem 2.** *Assume the sufficiently multiplicative differentiable function $g : \Omega \subseteq \mathbb{R} \to \mathbb{R}^+$ with $r_1$ multiplicative root in an open interval $\Omega$. Whenever $x_0$ is sufficiently close to $r_1$, the multiplicative Schröder scheme (7) has quadratic convergence.*

**Proof.** Let $r_1$ be a multiplicative root of function $g(x)$ such that $g(r_1) = 1$. Since the function $g(x)$ is sufficiently multiplicative differentiable, by using Equation (5) and the error equation $e_k = r_1 - x_k$, we have

$$g(r_1) = 1 = g(x_k)g^*(x_k)^{e_k}g^{**}(x_k)^{\frac{e_k^2}{2}}g^{***}(c_1)^{\frac{e_k^3}{6}}, \tag{8}$$

$$g(r_1) = 1 = g(x_k)g^*(x_k)^{e_k}g^{**}(c_2)^{\frac{e_k^2}{2}}, \tag{9}$$

where $c_1, c_2$ are between $r_1$ and $x_k$. Now, raising the power of (8) by $ln(g^*(x_k))$ gives

$$1 = g(x_k)^{ln(g^*(x_k))}g^*(x_k)^{ln(g^*(x_k))e_k}g^{**}(x_k)^{\frac{ln(g^*(x_k))}{2}e_k^2}g^{***}(c_1)^{\frac{ln(g^*(x_k))}{6}e_k^3}, \tag{10}$$

and raising the power of (9) by $e_k\, ln(g^{**}(x_k))$ gives

$$1 = g(x_k)^{e_k ln(g^{**}(x_k))}g^*(x_k)^{ln(g^{**}(x_k))e_k^2}g^{**}(c_2)^{ln(g^{**}(x_k))\frac{e_k^3}{2}}. \tag{11}$$

Dividing (10) by (11) gives

$$g(x_k)^{ln(g^*(x_k))} \left( \frac{g^*(x_k)^{ln(g^*(x_k))}}{g(x_k)^{ln(g^{**}(x_k))}} \right)^{e_k} \left( \frac{g^{**}(x_k)^{\frac{ln(g^*(x_k))}{2}}}{g^*(x_k)^{ln(g^{**}(x_k))}} \right)^{e_k^2} \left( \frac{g^{***}(c_1)^{\frac{ln(g^*(x_k))}{6}}}{g(c_2)^{\frac{ln(g^{**}(x_k))}{2}}} \right)^{e_k^3} = 1. \tag{12}$$

After using the natural log on both sides of (12) and the properties of the natural log, one can have

$$ln(g^*(x_k))ln(g(x_k)) + ln\left(\frac{g^*(x_k)^{ln(g^*(x_k))}}{g(x_k)^{ln(g^{**}(x_k))}}\right)e_k + ln\left(\frac{g^{**}(x_k)^{\frac{ln(g^*(x_k))}{2}}}{g^*(x_k)^{ln(g^{**}(x_k))}}\right)e_k^2 + O(e_k^3) = 0,$$

$$ln(g^*(x_k))ln(g(x_k)) + \left(ln\left(g^*(x_k)^{ln(g^*(x_k))}\right) - ln\left(g(x_k)^{ln(g^{**}(x_k))}\right)\right)e_k+$$

$$\left(ln\left(g^{**}(x_k)^{\frac{ln(g^*(x_k))}{2}}\right) - ln\left(g^*(x_k)^{ln(g^{**}(x_k))}\right)\right)e_k^2 + O(e_k^3) = 0,$$

$$ln(g^*(x_k))ln(g(x_k)) + \left((ln(g^*(x_k)))^2 - ln(g(x_k))ln(g^{**}(x_k))\right)e_k-$$

$$(ln(g^*(x_k))ln(g^{**}(x_k)))\frac{e_k^2}{2} + O(e_k^3) = 0. \tag{13}$$

Rearranging the terms of the Equation (13), we have

$$\frac{ln(g(x_k))ln(g^*(x_k))}{(ln(g^*(x_k)))^2 - ln(g(x_k))ln(g^{**}(x_k))} = -e_k$$

$$+ \frac{e_k^2}{2}\left(\frac{ln(g^*(x_k))ln(g^{**}(x_k))}{ln(g^*(x_k))^2 - ln(g(x_k))ln(g^{**}(x_k))}\right) + O(e_k^3). \tag{14}$$

Now, using $e_k = r_1 - x_k$ and the root $r_1$ on both sides of the Equation (7), we obtain

$$r_1 - x_{k+1} = r_1 - x_k + \frac{ln(g(x_k))ln(g^*(x_k))}{(ln(g^*(x_k)))^2 - ln(g(x_k))ln(g^*(x_k))},$$

$$e_{k+1} = e_k - e_k + e_k^2(B) + O(e_k^3),$$

$$e_{k+1} = e_k^2(B) + O(e_k^3), \tag{15}$$

where $B = \frac{1}{2}\left(\frac{ln(g^*(x_k))ln(g^{**}(x_k))}{ln(g^*(x_k))^2 - ln(g(x_k))ln(g^{**}(x_k))}\right)$.

Hence, technique (7) has quadratic convergence. □

## 4. Experimental Work

In this section, some experiments are performed on our iterative method and compared with the existing methods of similar order of convergence. We contrast our multiplicative Schröder method (*MSM*) to the well-known classical Schröder method (*SM*) (6). In addition, we also compare it with the modified Newton's method (*MNM*) [21], which is defined as

$$x_{k+1} = x_k - m\frac{g(x_k)}{g'(x_k)}. \tag{16}$$

The method (16) requires prior knowledge of multiplicity *m* of the required root. All the numerical work has been conducted using *Mathematica* 11. For the ordinary derivative case, the stopping criterion is $|g(x_k)| < 10^{-50}$, and in the multiplicative derivative case, $|g_1(x_k) - 1| < 10^{-50}$. The iteration index $k$, CPU timing, and consecutive iteration error $|x_{k+1} - x_k|$ are presented in Tables 1–6. Finally, the approximate computational order of convergence (ACOC) $\rho$ is calculated with the following formula:

$$\rho = \frac{ln\frac{|x_{p+1}-x_p|}{|x_p-x_{p-1}|}}{ln\frac{|x_p-x_{p-1}|}{|x_{p-1}-x_{p-2}|}}, \quad \text{for each } p = 2, 3, \ldots \tag{17}$$

**Remark 1.** *The meaning of expression $m(\pm n)$ is $m \times 10^{\pm n}$ in all the tables.*

**Example 1.** *Firstly, we consider the population growth model that formulates the following nonlinear function:*

$$g(x) = \frac{1000}{1564}e^x + \frac{435}{1564}(e^x - 1) - 1.$$

*In this model, we evaluate the birth rate denoted as x if a specific local area has 1000 thousand people at first and 435 thousand move into the local area in the first year. Likewise, we assume 1564 thousand individuals toward the finish of one year. The computed results towards the required zero $x_r = 0.1009979\ldots$ are displayed in Table 1. Clearly, the method MSM demonstrates better results in terms of consecutive error, number of iterations, and CPU timing in comparison with existing ones.*

**Table 1.** Convergence behavior of the methods *MNM*, *SM*, and *MSM* at approximation $x_0 = 1$.

| Schemes | $k$ | $\lvert x_{(k+1)} - x_{(k)} \rvert$ | $\rho$ | Total Number of Iterations | CPU Time (Seconds) |
|---|---|---|---|---|---|
| | 2 | $3.7(-2)$ | 2.000 | 5 | 0.203 |
| MNM | 3 | $6.7(-4)$ | | | |
| | 4 | $2.1(-7)$ | | | |
| | 2 | $1.1(-1)$ | 2.006 | 6 | 0.157 |
| SM | 3 | $5.8(-3)$ | | | |
| | 4 | $1.6(-5)$ | | | |
| | 2 | $1.2(-5)$ | 2.001 | 4 | 0.062 |
| MSM | 3 | $6.4(-12)$ | | | |
| | 4 | $1.8(-24)$ | | | |

**Example 2.** *Here, we study the nonlinear problem $g(x) = xe^{x^2} - \sin^2(x) + 3\cos(x) - 4$ having a zero $x_r = 1.06513\ldots$. The evaluated results are demonstrated in Table 2. From the obtained results in Table 2, we can say that our method is faster than the existing methods since our scheme converges to the required root in only four iterations compared with the others that required seven and eight. In addition, our scheme has the lowest absolute error difference and CPU timing among the mentioned methods.*

**Table 2.** Convergence behavior of the methods *MNM*, *SM*, and *MSM* at approximation $x_0 = 0.75$.

| Schemes | $k$ | $\lvert x_{(k+1)} - x_{(k)} \rvert$ | $\rho$ | Total Number of Iterations | CPU Time (Seconds) |
|---|---|---|---|---|---|
| | 2 | $2.1(-1)$ | 2.938 | 8 | 0.078 |
| MNM | 3 | $2.1(-1)$ | | | |
| | 4 | $1.9(-1)$ | | | |
| | 2 | $9.1(-2)$ | 2.000 | 7 | 0.109 |
| SM | 3 | $4.3(-2)$ | | | |
| | 4 | $6.0(-3)$ | | | |
| | 2 | $3.7(-3)$ | 1.998 | 4 | 0.062 |
| MSM | 3 | $1.1(-5)$ | | | |
| | 4 | $8.7(-11)$ | | | |

**Example 3.** *Now, we test the methods on the continuous stirred tank reactor problem, which was converted into the following mathematical expression by Douglas [22]:*

$$\kappa \frac{2.98(s + 2.25)}{(s + 1.45)(s + 2.85)^2(s + 4.35)} = -1. \tag{18}$$

*Here, $\kappa$ denotes the gain of the proportional controller. For the values of $\kappa$, the control system is stable; however, when $\kappa = 0$, we have the poles of the open-loop transferred function as the solutions of the following nonlinear function:*

$$g(x) = x^4 + 1.5x^3 + 47.49x^2 + 83.06325x + 5.123266875. \tag{19}$$

The function $g(x)$ has zero $-2.85$ with multiplicity $m = 2$. The outcomes of the suggested method are demonstrated in Table 3 and results are equally competent compared with those of MNM and SM.

**Table 3.** Convergence behavior of the methods *MNM*, *SM*, and *MSM* at approximation $x_0 = -2.5$.

| Schemes | $k$ | $\lvert x_{(k+1)} - x_{(k)} \rvert$ | $\rho$ | Total Number of Iterations | CPU Time (Seconds) |
|---|---|---|---|---|---|
| MNM | 2 | 8.0(−6) | 1.903 | 4 | 0.093 |
| | 3 | 1.5(−12) | | | |
| | 4 | 5.6(−26) | | | |
| SM | 2 | 1.6(−4) | 2.187 | 4 | 0.078 |
| | 3 | 5.8(−10) | | | |
| | 4 | 8.1(−21) | | | |
| MSM | 2 | 3.4(−4) | 2.266 | 4 | 0.125 |
| | 3 | 2.7(−9) | | | |
| | 4 | 1.7(−19) | | | |

**Example 4.** *Lastly, we worked on the Van der Waals equation of ideal gas [23], which describes the characteristics of real gas, and formed it into the following mathematical expression:*

$$g(x) = x^3 - 5.22x^2 + 9.0825x - 5.2675.$$

*One of its zeros, $x_r = 1.75$, has multiplicity $m = 2$. The performance of different iterative schemes has been shown in Table 4 and one can easily conclude that the proposed method MSM converges much faster to the root than the other methods MNM and SM.*

**Table 4.** Convergence behavior of the methods *MNM*, *SM*, and *MSM* at approximation $x_0 = 1.9$.

| Schemes | $k$ | $\lvert x_{(k+1)} - x_{(k)} \rvert$ | $\rho$ | Total Number of Iterations | CPU Time (Seconds) |
|---|---|---|---|---|---|
| MNM | 2 | 9.0(−3) | 1.995 | 6 | 0.078 |
| | 3 | 1.1(−3) | | | |
| | 4 | 2.0(−5) | | | |
| SM | 2 | 1.7(−3) | 2.050 | 5 | 0.094 |
| | 3 | 5.4(−5) | | | |
| | 4 | 5.0(−8) | | | |
| MSM | 2 | 1.2(−3) | 2.033 | 5 | 0.062 |
| | 3 | 2.7(−5) | | | |
| | 4 | 1.3(−8) | | | |

**Example 5.** *Eigenvalues play a significant role in linear algebra and in many applications of image processing. However, it is sometimes a tough task to evaluate eigenvalues if we have a matrix of larger size. So, here, we focus on finding the eigenvalues of the following ninth-order matrix:*

$$B = \frac{1}{8}\begin{pmatrix} -12 & 0 & 0 & 19 & -19 & 76 & -19 & 18 & 437 \\ -64 & 24 & 0 & -24 & 24 & 64 & -8 & 32 & 376 \\ -16 & 0 & 24 & 4 & -4 & 16 & -4 & 8 & 92 \\ -40 & 0 & 0 & -10 & 50 & 40 & 2 & 20 & 242 \\ -4 & 0 & 0 & -1 & 41 & 4 & 1 & 2 & 25 \\ -40 & 0 & 0 & 18 & -18 & 104 & -18 & 20 & 462 \\ -84 & 0 & 0 & -29 & 29 & 84 & 21 & 42 & 501 \\ 16 & 0 & 0 & -4 & 4 & -16 & 4 & 16 & -92 \\ 0 & 0 & 0 & 0 & 0 & 0 & 0 & 0 & 24 \end{pmatrix},$$

*The characteristic equation of matrix B forms the following polynomial function:*

$$g_5(x) = x(x^8 - 29x^7 + 349x^6 - 2261x^5 + 8455x^4 - 17663x^3 + 15927x^2 + 6993x - 24732) + 12960.$$

*This function has a zero $x = 3$ of multiplicity $m = 4$. Table 5 reports the results of the proposed scheme, which are much better in contrast with the available techniques in terms of errors, order of convergence, and CPU time. Further, no doubt, MSM consumes an equal number of iterations but with the lowest CPU time and less error.*

**Table 5.** Convergence behavior of the methods *MNM*, *SM*, and *MSM* at approximation $x_0 = \frac{31}{10}$.

| Schemes | $k$ | $\|x_{(k+1)} - x_{(k)}\|$ | $\rho$ | Total Number of Iterations | CPU Time (Seconds) |
|---------|-----|--------------------------|--------|----------------------------|--------------------|
| *MNM*   | 2   | 2.0(−6)                  | 2.000  | 4                          | 0.156              |
|         | 3   | 9.2(−13)                 |        |                            |                    |
|         | 4   | 2.0(−25)                 |        |                            |                    |
| *SM*    | 2   | 2.5(−6)                  | 2.000  | 4                          | 0.163              |
|         | 3   | 1.5(−12)                 |        |                            |                    |
|         | 4   | 5.5(−25)                 |        |                            |                    |
| *MSM*   | 2   | 8.7(−7)                  | 2.000  | 4                          | 0.125              |
|         | 3   | 1.8(−13)                 |        |                            |                    |
|         | 4   | 7.6(−27)                 |        |                            |                    |

**Example 6.** *Lastly, we applied the proposed methods to the clustering problem defined as*

$$g_6(x) = (x-1)^{120}(x-2)^{150}(x-3)^{100}(x-4)^{55}.$$

*The function $g_6(x)$ has the zeros $1, 2, 3, 4$ with multiplicity $120, 150, 100, 55$, respectively. In this example, we approximated the zero $1$ with multiplicity $120$. In Table 6, the numerical results are depicted.*

**Table 6.** Convergence behavior of the methods *MNM*, *SM*, and *MSM* at approximation $x_0 = \frac{9}{10}$.

| Schemes | $k$ | $\|x_{(k+1)} - x_{(k)}\|$ | $\rho$ | Total Number of Iterations | CPU Time (Seconds) |
|---------|-----|--------------------------|--------|----------------------------|--------------------|
| *MNM*   | 2   | 3.6(−4)                  | 2.000  | 4                          | 0.093              |
|         | 3   | 2.4(−7)                  |        |                            |                    |
|         | 4   | 1.0(−13)                 |        |                            |                    |
| *SM*    | 2   | 4.4(−4)                  | 2.000  | 4                          | 0.188              |
|         | 3   | 3.5(−7)                  |        |                            |                    |
|         | 4   | 2.2(−13)                 |        |                            |                    |
| *MSM*   | 2   | 4.4(−4)                  | 2.000  | 4                          | 0.156              |
|         | 3   | 3.5(−7)                  |        |                            |                    |
|         | 4   | 2.2(−13)                 |        |                            |                    |

**Remark 2.** *The graphical error analysis of Examples 1 to 6 is shown in Figure 1. It is clear from all subfigures of Figure 1 that our method of error reduction is faster than existing methods. In a similar way, iteration comparisons of different existing methods with our method are given in Figure 2. Clearly, the proposed method converges to root in less iterations compared with other schemes.*

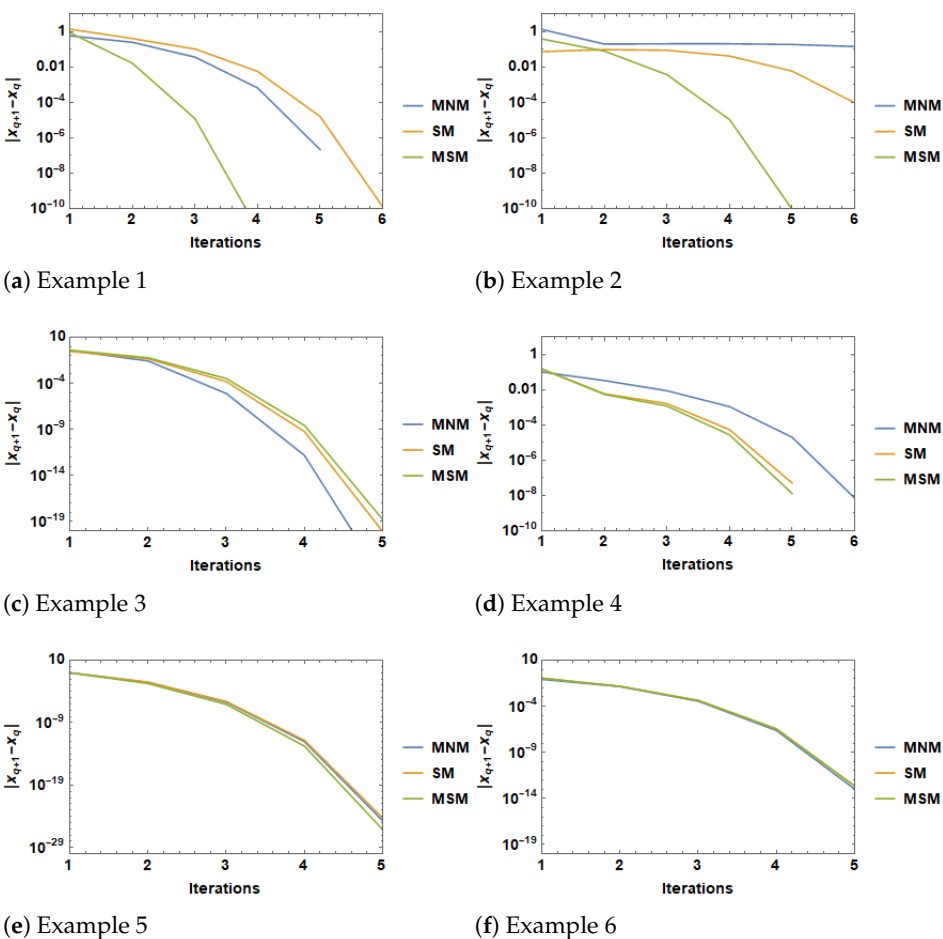

**Figure 1.** Graphical error analysis.

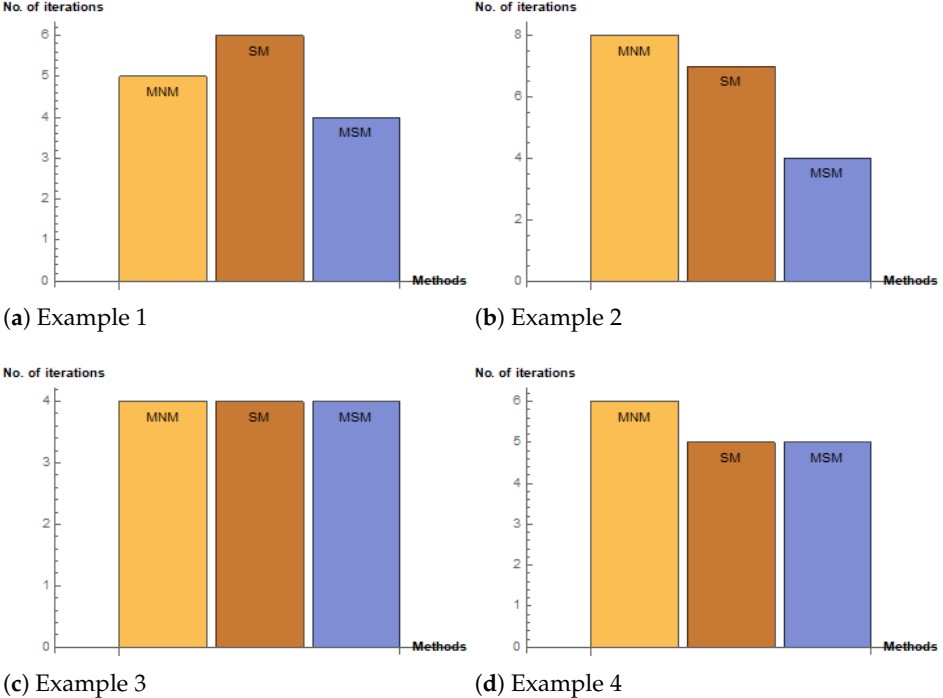

**Figure 2.** *Cont.*

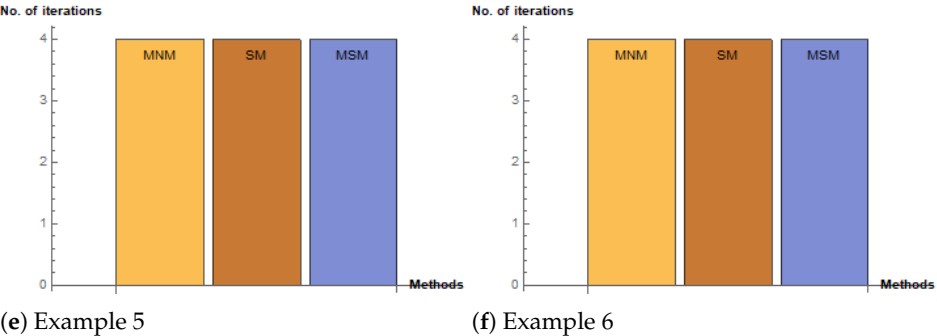

**(e)** Example 5          **(f)** Example 6

**Figure 2.** Iteration analysis.

## 5. Basin of Attraction

The concept of the basin of attraction confirms the convergence of all the possible roots of the nonlinear equation within a specified rectangular region. So, we also present dynamical planes [24] of modified Newton's method ($MNM$), ordinary Schröder method ($SM$), and multiplicative Schröder method ($MSM$) on different initial values in the rectangular region $[-2.5, 2.5] \times [-2.5, 2.5]$. We have chosen three problems to analyze the basin of attraction for comparison of these three methods. Each image is plotted by an initial guess as an ordered pair of 256 complex points of the abscissa and coordinate axis. If an initial point does not converge to the required root, it is plotted with black color; otherwise, different colors are used to represent different roots with tolerance $10^{-3}$.

**Example 7.** *The scalar function $z^2 - 1$ has the zeros $\{-1, 1\}$. In Figure 3, pink and yellow colors represent the convergence of zeros and black color for the divergence. It is clear that the proposed methods are approaching the desired zero.*

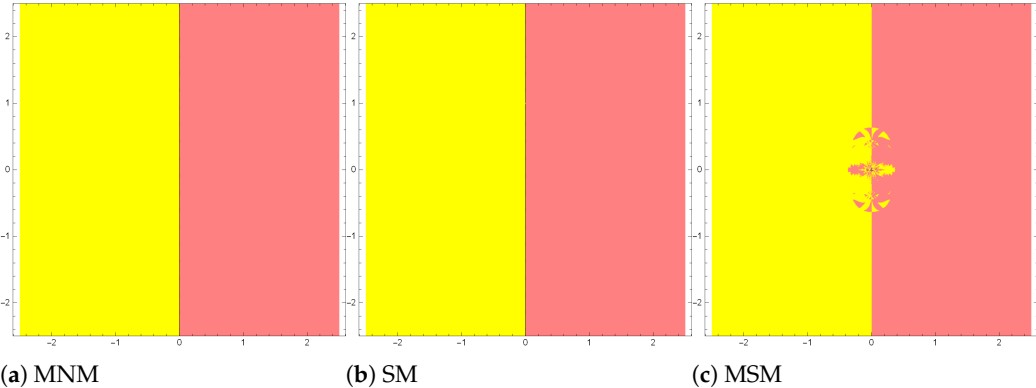

**(a)** MNM                    **(b)** SM                    **(c)** MSM

**Figure 3.** Dynamical planes of new and existing methods for Example 7.

**Example 8.** *The nonlinear function $z^3 - 1$, having the zeros $\{1, e^{\frac{2\pi i}{3}}, e^{\frac{4\pi i}{3}}\}$, is tested and the basin of attraction is shown in Figure 4. The divergence area is very small in MSM.*

**Example 9.** *Lastly, the basin of attraction of the nonlinear function $z^3 + z$ with zeros $\{0, -i, i\}$ is shown in Figure 5. It is clear that the method SM has a more divergent area in comparison with the proposed method.*

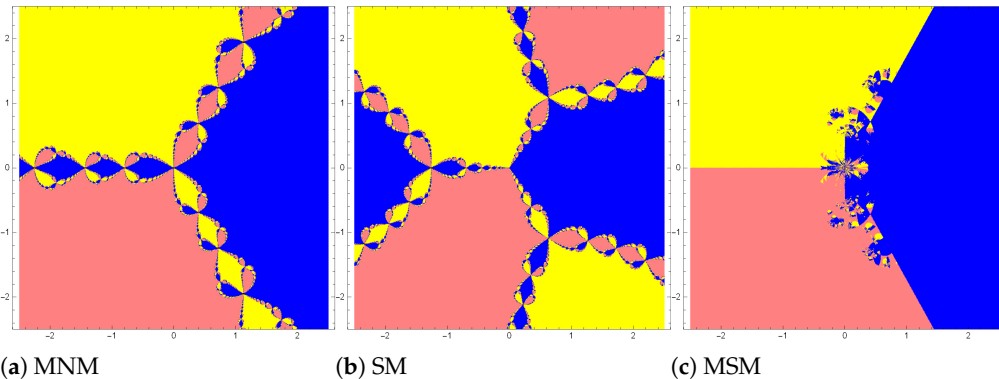

(**a**) MNM  (**b**) SM  (**c**) MSM

**Figure 4.** Dynamical planes of new and existing methods for Example 8.

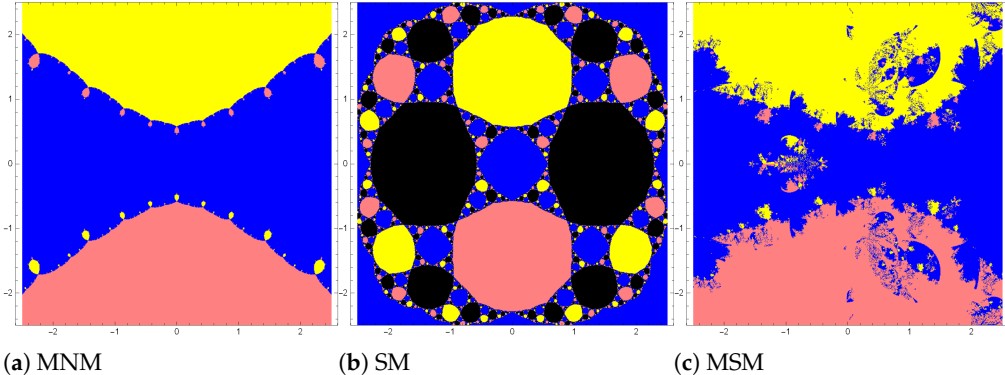

(**a**) MNM  (**b**) SM  (**c**) MSM

**Figure 5.** Dynamical planes of new and existing methods for Example 9.

## 6. Conclusions

In this paper, we proposed a new iterative method with the help of MCA. Schröder's iterative method and multiplicative derivatives are the two main pillars of our scheme. We studied the convergence analysis of the presented method. The suggested scheme did not require the prior knowledge of multiplicity *m*. In addition, we also provide a more efficient solution to the population growth, continuous stirred tank reactor (CSTR), ideal gas, and academic problems compared with the existing solutions.

We compared our techniques on the basis of (i) the absolute error difference between two consecutive iterations, (ii) the order of convergence, (iii) the number of iterations, (iv) CPU timing, (v) the graphs of absolute errors, and (vi) bar graphs. In all six different ways, we found that our method performs much better in comparison with the existing methods. Finally, we studied the basin of attraction, the findings of which also support the numerical results. In future work, we will focus on the multi-point iterative methods for multiple roots as well as for systems of nonlinear equations. This area will open a new, veritable Pandora's Box of iterative methods.

**Author Contributions:** Conceptualization, S.B. and R.B.; Methodology, G.S., S.B. and R.B.; Software, S.B. and R.B.; Validation, G.S.; Writing—original draft, G.S., S.B. and R.B.; Supervision, S.B. and R.B. All authors have read and agreed to the published version of the manuscript.

**Funding:** The Deanship of Scientific Research (DSR) at King Abdulaziz University, Jeddah, Saudi Arabia has funded this project under grant no.(KEP-MSc-58-130-1443).

**Acknowledgments:** The Deanship of Scientific Research (DSR) at King Abdulaziz University, Jeddah, Saudi Arabia has funded this project under grant no.(KEP-MSc-58-130-1443). The authors, therefore, acknowledge with thanks DSR for technical and financial support.

**Conflicts of Interest:** The authors declare no conflict of interest.

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
