# Peer review of "A Multiplicative Calculus Approach to Solve Applied Nonlinear Models"

_mca, doi:10.3390/mca28020028_

Round 1
Reviewer 1 Report
The authors claim to propose a new iterative scheme for multiple roots without prior knowledge of the multiplicity by adopting multiplicative calculus instead of standard calculus. The new scheme is based on the well-known Schroder method, with the order of convergence extended from the second to the fourth.
The work leaves an impression of frivolous written and presented content. The exposition of the material is in loose form and in an incomplete presentation.
For example, in Section 1 a theorem is formulated (Theorem 1) without citing a source.
Section 2. "Proposed Schemes with some basic terminologies" is too short and lacks sufficient explanation on obtaining the new schemes.
I believe that the ordinary Schroder method should be briefly included here and the method of obtaining the new schemes for the completeness of the section should be clarified.
Section 3 lacks any explanatory text, along with the formulation and proof of the two theorems.
I noticed a lot of minor mistakes, such as the following:
1. in Definition 1: text "Suppose g ... be a positive nonlinear equation" should be "Suppose g ... be a positive nonlinear function"
2. in Theorem 1: text "..(n + 1) times multiplicative differential.." should be "...(n + 1) times multiplicative differentiable..."
3. in Theorem 2: text "Assume the sufficiently multiplicative differential function g ..." should be "Assume the sufficiently multiplicative differentiable function g"
4. Need an explanation for Theorem 2, whether you formulate and prove it or it is formulated and proved in other sources.
5. line 80: text "Hence, technique (2.1) has quadratic convergence" should be "Hence, technique (6) ..."
6. line 84: text "Proof. Proof:" should be "Proof."
7. Example 4. "... van der waal equation..." You must use capital letters
8.The formula of the mentioned modified Newton's method in Section 4. "Experimental work" should be formulated
9. line 125: ".. scaler equation .."
Among the main shortcomings of the manuscript are the following:
1. When comparing different iteration schemes, it is not enough to just examine the error, exec.time and the number of iterations. It is appropriate to add an analysis of the computational efficiency of the proposed schemes and a comparative analysis with the known methods.
2. From the examples presented, it can be seen that modified Newton's method (MNM) gives better CPU time in all examples except the last example.
3. The article states that the new schemes are for multiple roots, without prior knowledge of multiplicity, but only two examples are presented where the roots are where the roots are with multiplicity two. There are no examples with roots of greater multiplicity.
The manuscript requires thorough and major revision. It requires a lot of serious work in order to improve it and bring it into a form suitable for publication.
Author Response
Please find the reply to the reviewers in the attached file.

Author Response
Please find the attached file of replies to reviewers' comments.

Round 2
Reviewer 1 Report
In its current form, the manuscript is already in a much more acceptable form compared to its original form.
I noticed a few small bugs:
1) Example 4: " ... equation z^2 − 1 ..." should be "... equation z^2 − 1=0 ..."
or you should use the term "function" instead of "equation".
Similarly for Examples 8, 9 and 3.
2) In Example 7: "scaler" instead of "scalar"
3) In Example 8: "... equation z^3 − 1 having the roots {1, −i, i} ..."
Actually "−i" and "i" are not zeros of equation z^3 − 1=0.
After correcting the mentioned remarks, the work can be published.
Author Response
1) Example 4: " ... equation z^2 − 1 ..." should be "... equation z^2 − 1=0 ..."
or you should use the term "function" instead of "equation".
Similarly for Examples 8, 9 and 3.
Reply: The suggestion is incorporated in the revised draft.
2) In Example 7: "scaler" instead of "scalar"
Reply: The typo is fixed.
3) In Example 8: "... equation z^3 − 1 having the roots {1, −i, i} ..."
Actually "−i" and "i" are not zeros of equation z^3 − 1=0.
Reply: We fixed the typo. The roots are rectified in the revised draft.
Reviewer 2 Report
1. Line 118: Generally tolerance value should be very small. However, you have taken a very very large tolerance. I wonder how this value will work.
2. No attention was paid to improve English. There are several grammatical mistakes which must be corrected.
Author Response
- Line 118: Generally tolerance value should be very small. However, you have taken a very very large tolerance. I wonder how this value will work.
Reply: It is just typing mistake, we have taken tolerance 10-3.
- No attention was paid to improve English. There are several grammatical mistakes which must be corrected.
Reply: Please see the revised draft.
Round 3
Reviewer 2 Report
The authors have made substantial improvements to the paper. So, I recommend the paper for publication.